# Efficacy and safety of Weifuchun tablet for chronic atrophic gastritis: A systematic review and meta-analysis

**Longhua Wang**[1], **Xia Ding**[2]\*, **Ping Li**[1], **Fuwen Zhang**[1], **Shuying Ru**[1], **Fenglei Wang**[1], **Lan Li**[3]

**1** Department of Gastroenterology, Dongzhimen Hospital, Beijing University of Chinese Medicine, Beijing, China, **2** School of Traditional Chinese Medicine, Beijing University of Chinese Medicine, Beijing, China, **3** Out-patient Department, Dongzhimen Hospital, Beijing University of Chinese Medicine, Beijing, China

☯ These authors contributed equally to this work.
\* dingx@bucm.edu.cn

**Data Availability Statement:** All relevant data are within the paper and its Supporting Information files. We also uploaded our dataset on Figshare (DOI: 10.6084/m9.figshare.22298005.v2).

## Abstract

### Background

Chronic atrophic gastritis is a significant premalignant lesion of gastric carcinoma. There is a great need to prevent the progression to gastric carcinoma through early intervention and treatment for chronic atrophic gastritis. Weifuchun, a famous Chinese patent drug, has been widely used for chronic atrophic gastritis in China. However, it remains unclear whether Weifuchun is effective for atrophic gastritis.

### Objective

To determine the effectiveness and safety of Weifuchun for chronic atrophic gastritis.

### Methods

We systematically retrieved seven databases (Cochrane Library, EMBASE, PubMed, China National Knowledge Infrastructure, Wanfang database, Chinese Scientific Journals Database, and Chinese Biological Medical Database) from their inception to October 5, 2022. Methodological quality was examined using the Cochrane Risk of bias tool. We also used RevMan 5.4 software for statistical analysis to examine the effectiveness and safety of Weifuchun.

### Results

Fifteen studies with 1,488 patients were enrolled in this meta-analysis. The study indicated that Weifuchun was more effective (RR 1.52; 95% CI 1.41, 1.63; p<0.00001) than Western medicine and other Chinese patent medicine. In addition, Weifuchun was more effective in improving gastric mucosal under gastroscopy, improving histopathologic changes of gastric mucosal, and inhibiting Helicobacter pylori. However, no significant difference in safety was examined between Weifuchun and the control group (RR 2.83; 95% CI 0.85, 9.38; P = 0.09).

**Funding:** The author(s) received no specific funding for this work.

**Competing interests:** The authors have declared that no competing interests exist.

## Conclusions

The meta-analysis revealed a significant statistical difference with Weifuchun in effectiveness compared to the control group. However, there was no significant difference in safety. Thus, more high-quality clinical studies are needed in the future.

## Trial registration

Registration number CRD42022365703.

## Introduction

Gastric carcinoma (GC) is one of the most common malignant tumors worldwide, with high morbidity and mortality [1]. However, there is great significance for five years overall survival rate with early intervention and treatment for gastric malignancies [2]. Occurrences of GC undergo the following sequential stages: chronic gastritis, atrophic gastritis, intestinal metaplasia (IM), dysplasia, and carcinoma, known as the Correa model [3–5]. Chronic atrophic gastritis (CAG), characterized by a decrease in intrinsic glands of gastric mucosa [6], is now defined as a critical premalignant lesion of gastric cancer (PLGC) [7]. Furthermore, a multicenter study in China has shown that the proportion of CAG in patients with gastritis reached 25.8% [8]. Thus, sufficient attention to managing CAG is crucial to prevent progression to GC.

Helicobacter pylori (H. Pylori), acknowledged as a class I carcinogenic factor, is regarded as an influential cause of CAG [9, 10]. To reduce GC risk, H. pylori eradication is strongly recommended with gastritis [11]. Nevertheless, a growing body of research has indicated that antimicrobial resistance and adverse drug events have markedly increased [12, 13]. The effectiveness and safety of conventional treatment remain insufficient, and it is of great significance to search replacement therapies to relief symptoms and improve life quality in CAG patients [14].

Chinese herbal medicine is widely applied to treat CAG based on its multiple targets and few side effects [15]. Weifuchun (WFC) tablet, a well-known Chinese patent medicine, is consisted of three herbs, Renshen (Red ginseng), Xiangchacai (Isodon amethystoides), and Zhike (Fructus Aurantii). Importantly, WFC was approved by China Food and Drug Administration (CDFA) in 1982. The function of WFC is to strengthen the spleen and replenish qi, promote blood circulation and detoxification, and eliminate gas and phlegm [16]. It has been commonly used in therapy for chronic gastritis and PLGC [17, 18]. According to the 2017 consensus on integrated traditional Chinese and Western medicine for CAG, WFC could be a replacement treatment for CAG [19]. Nevertheless, it remains unclear whether WFC is effective for treating CAG. Thus, it is of great significance and necessity to systematically evaluate the clinical effectiveness of WFC. Thus, we conducted a meta-analysis to determine its effectiveness and safety, aimed at providing available evidence for clinical treatment.

## Materials and methods

### Search strategy

This study was enrolled at PROSPERO (registration number: CRD42022365703; http://www.crd.york.ac.uk/prospero). Databases, including Cochrane Library, EMBASE, PubMed, China National Knowledge Infrastructure (CNKI), Wanfang database, Chinese Scientific Journals Database (VIP) and Chinese Biological Medical Database (CBM), were searched from their inception to October 5, 2022. Additionally, we searched the Chinese Clinical Trial Registry

and ClinicalTrials.gov. Two researchers (Longhua Wang and Ping Li) conducted and screened all the citations independently. Furthermore, we conducted a retrieval combined Mesh terms with free words, and a complete list of retrieval strategies for PubMed was described below:

#1. gastritis, atrophic [Mesh Terms]

#2. ((((((' Atrophic Gastriti* '[Title/Abstract]) OR ('precancerous conditions'[Title/Abstract])) OR (Metaplasia [Title/Abstract])) OR ('Gastric premalignant'[Title/Abstract])) OR ('Intestin* metaplasia'[Title/Abstract])) OR (Dysplasia [Title/Abstract])) OR ('gastric atrophy'[Title/Abstract])

#3. (Weifuchun [Title/Abstract]) OR (WFC[Title/Abstract])

#4. #1 OR #2

#5. #4 AND #3

#6. (((('Randomized Controlled Trial'[Publication Type]) OR ('Controlled Clinical Trial'[Publication Type])) OR (Randomized [Title/Abstract])) OR (Placebo [Title/Abstract])) OR (Randomly [Title/Abstract])

#7. #5 AND #6

## Selection criteria

The retrieved results were imported into EndNote X9.1. Two researchers (Longhua Wang and Ping Li) independently evaluated the qualification of retrieval studies according to inclusion criteria. A preliminary screening was performed by independently browsing titles and abstracts. Based on browsing titles and abstracts, some studies were excluded including case reports, experience introductions, reviews, animal studies, irrelevant to our topic and non-randomized controlled trials. We also excluded studies that combined WFC with other therapies. A next step screening was conducted, and the study was temporarily adopted if there was no clear exclusion information.

Two researchers (Longhua Wang and Ping Li) independently performed a next- step screening by browsing the full texts. Based on reading the full article, we screened the selected studies according to inclusion criteria. We would discuss divergences with a third researcher (Xia Ding) if they existed.

Studies were included according to the inclusion criteria as follows. 1) The studies were randomized controlled trials (RCTs) published in Chinese or English before October 5, 2022, and no restriction on the implementation of blinding. 2) The study subjects were adult patients with CAG diagnosed by endoscopy and pathology. The gender and source of the cases were not limited. 3) The experiment group was treated with the WFC tablet alone, but interventions of the control group included Western medicine, Chinese herbal compound, Chinese patent drugs, or a placebo.4) Duration of the course was 3 months at a minimum.

## Data extraction and quality assessment

Data extraction was done independently by two investigators (Longhua Wang and Lan Li), containing the literature title, the first author, date of publication, study population and baseline data consistency, sample size, intervention measures, duration, outcome evaluation indicators and results, follow-up time and adverse drug events. If divergences existed, we would discuss them with a third researcher (Shuying Ru).

The methodological quality of RCTs was assessed using the Cochrane risk of bias tool, containing seven criteria (random sequence generation, allocation concealment, blinding of participants and personnel, blinding of outcome assessment, incomplete outcome data, selective reporting, and other biases). Two researchers (Longhua Wang and Fenglei Wang) conducted a standalone methodological quality assessment, and gave a bias assessment graded as either

low, high, or unknown risk. Furthermore, inconsistencies were resolved by discussion with all researchers.

## Data synthesis and analysis

The RevMan 5.4 software was for statistical analysis. Dichotomous data was assessed by relative risks (RR) and 95% confidence intervals (CI). Additionally, continuous variable data was evaluated by standardized mean difference (SMD) and 95% CI. The $\chi^2$ test and inconsistency index statistic($I^2$) were also used to examine heterogeneity [20]. We conducted a random effects model to examine pooled RR if significant heterogeneity existed ($I^2 > 50\%$ or $P < 0.05$). If not, a fixed effects model was conducted. We conducted a sensitivity analysis to examine sources of heterogeneity and the stability of results. Furthermore, we used a Z test for the overall effect. Pooled results were considered statistically significant when $p < 0.05$. Moreso, funnel plots were performed to evaluate potential publication bias.

## Results

### Description of studies

According to the search strategy, 2,844 studies were adopted, including 591 in CNKI, 943 in the Wanfang database, 310 in VIP, 959 in CBM, 7 in PubMed, 16 in Embase, and 18 in Cochrane Library. Forty-six studies were enrolled after the first screening. Studies including non-RCTs, incomplete date, and not accordant with the inclusion criteria were eliminated following the next step of screening. As a result, 15 RCTs were enrolled in the meta-analysis [21–35]. A flow chart (**Fig 1**) describes how the studies were selected. Furthermore, the sample size was between 35 and 248. The duration of intervention was 3 to 12 months. The interventions were WFC compared with Western medicine, placebo, and Chinese patent drugs. The principal characteristics of enrolled studies in this meta-analysis are described in **Table 1**.

### Risk of bias assessment

The risk of bias was examined using the Cochrane Risk of bias tool, and the methodological quality assessment of all enrolled studies is described in **Table 2**. Three studies [29, 32, 35] used the method of random number tables. Yanqin Bian [34] used the method of Computer-generated randomization, but the other studies [21–28, 30, 31, 33] just described "randomization". Three studies reported "double-blind" methods [25, 31, 34], and the study of Yanqin Bian [34] was the only one that elaborated on how to blind and allocate concealment. Two studies described cases of lost to follow-up [29, 34], but the lost data was not examined by intention to treat analysis. Research design limitations made it difficult to determine whether randomization, blinding, or allocation concealment was executed adequately, leading to a high risk of bias in all studies. Evaluation of the risk of bias is detailed in **Fig 2**.

### Outcomes

**Comparison of clinical efficacy.** Fourteen studies [21–33, 35] reported the effective clinical rate. A fixed effects model was conducted to analyze heterogeneity analysis results ($P = 0.09$, $I^2 = 36\%$), indicating no significant heterogeneity among fourteen studies. The differences between WFC and the control group were statistically significant (RR 1.52; 95% CI 1.41, 1.63; $P < 0.00001$). The overall effects($Z = 11.58$, $p < 0.00001$) showed that WFC was more effective in clinical efficacy than the control group (**Fig 3**).

**Sensitivity analysis.** According to the result of heterogeneity ($P = 0.09$, $I^2 = 36\%$), we performed sensitivity analysis to assess the stability of outcomes and underlying sources of

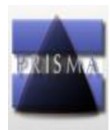

PRISMA 2009 Flow Diagram

```
Identification

    Records identified through              Additional records identified
    database searching                      through other sources
    (n = 2844)                              (n = 0)

                        Records after duplicates removed
                        (n = 1429)

Screening

                        Records screened                Records excluded
                        (n = 46)                         (n = 1383)

Eligibility

                        Full-text articles              Full-text articles
                        assessed for eligibility        excluded, with reasons
                        (n = 15)                         Not RCT (n=18)
                                                        Incomplete date (n=5)
                                                        Not accordant with the
                                                        inclusion criteria (n=8)

                        Studies included in
                        qualitative synthesis
                        (n = 15)

Included

                        Studies included in
                        quantitative synthesis
                        (meta-analysis)
                        (n = 15)
```

**Fig 1. Flow chart of study screening.**

**Table 1. Principal characteristics of all eligible studies.**

| Study (First Author, Year) | Sample Size (E/C) | Experiment group | Control group | Duration | Methodological characteristics | Outcome measures |
|---|---|---|---|---|---|---|
| Zhang YG, 2004 [25] | 32/26 | W, 4 tablets, TID | Vitacoenzyme, 4 tablets, TID | 6months | Randomized controlled | ABCE |
| Xu XF, 2018 [33] | 62/62 | W, 1.44g, TID | Folic acid tablet, 5mg, TID | 3months | Randomized controlled | ACE |
| Yuan LL, 2016 [31] | 42/38 | W, 4 tablets, TID | Placebo, 4tablets, TID | 9months | Randomized controlled | AE |
| Lin H, 2011 [29] | 60/60 | W, 1.436g, TID | Vitacoenzyme, 0.6-1g, TID | 6months | Randomized controlled | AE |
| Kong SL, 2018 [32] | 18/17 | W, 4 tablets, TID | Placebo, NR | 12months | Randomized controlled | ABD |
| Zhao H, 2004 [26] | 144/104 | W, 4 tablets, TID | Vitacoenzyme, 6 tablets, TID | 6months | Randomized controlled | ABD |
| Wang F, 2003 [23] | 40/40 | W, 4 tablets, TID | Folic acid tablet, 10mg, TID | 6months | Randomized controlled | A |
| Wu HM, 2000 [21] | 58/36 | W, 4 tablets, TID | Vitacoenzyme, 4 tablets, TID | 12months | Randomized controlled | ABCE |
| Lei YQ, 2010 [28] | 78/74 | W, 4 tablets, TID | Folic acid tablet, 10mg, TID | 3months | Randomized controlled | A |
| Zhao JY, 2003 [24] | 58/36 | W, 4 tablets, TID | Vitacoenzyme, 4 tablets, TID | 12months | Randomized controlled | AE |
| Zhu NG, 2001 [22] | 50/48 | W, 4 tablets, TID | Weisu granule, 5g, TID | 3months | Randomized controlled | AE |
| Gu C, 2016 [30] | 55/55 | W, 4 tablets, TID | Vitacoenzyme, 4 tablets, TID | 6months | Randomized controlled | AE |
| Wang HR, 2022 [35] | 38/38 | W, 4 tablets, TID | Omeprazole, 20mg, QD | 3months | Randomized controlled | A |
| Yang LL, 2006 [27] | 63/63 | W, 4 tablets, TID | Conventional therapy, NR | 3months | Randomized controlled | A |
| Bian YQ, 2021 [34] | 30/30 | W, 1.44g, TID | Vitacoenzyme, 0.8g, TID | 6months | Randomized controlled | C |

Annotation: W = Weifuchun; NR = not reported; A = Clinical effective rate; B = Improvement of gastric mucosa under gastroscopy; C = Histopathologic variations of gastric mucosa; D = H pylori inhibition rate; E = Adverse reactions

**Table 2. Methodological quality assessment of all eligible studies.**

| Study (First Author, Year) | Randomization | Allocation concealment | Blinding | Cases dropped out | Follow up | Adverse reactions |
|---|---|---|---|---|---|---|
| Zhang YG, 2004 [25] | Random | NR | Double-blind | No | NR | No |
| Xu XF, 2018 [33] | Random | NR | NR | No | NR | 5 cases in E and 2cases in C |
| Yuan LL, 2016 [31] | Random | NR | Double-blind | No | NR | 2 cases in E and 1 case in C |
| Lin H, 2011 [29] | Random number table | NR | NR | 3 cases in E and 4 cases in C | NR | No |
| Kong SL, 2018 [32] | Random number table | NR | NR | No | NR | NR |
| Zhao H, 2004 [26] | Random | NR | NR | No | NR | NR |
| Wang F, 2003 [23] | Random | NR | NR | No | NR | NR |
| Wu HM, 2000 [21] | Random | NR | NR | No | NR | 4 cases in E |
| Lei YQ, 2010 [28] | Random | NR | NR | No | NR | NR |
| Zhao JY, 2003 [24] | Random | NR | NR | No | NR | No |
| Zhu NG, 2001 [22] | Random | NR | NR | No | NR | No |
| Gu C, 2016 [30] | Random | NR | NR | No | NR | No |
| Wang HR, 2022 [35] | Random number table | NR | NR | No | NR | NR |
| Yang LL, 2006 [27] | Random | NR | NR | No | NR | NR |
| Bian YQ, 2021 [34] | Computer-generated randomization | Allocation concealment by computer | Double-blind | 6 cases in E and 4 cases in C | NR | NR |

Annotation: NR = not reported, E = experiment group, C = control group.

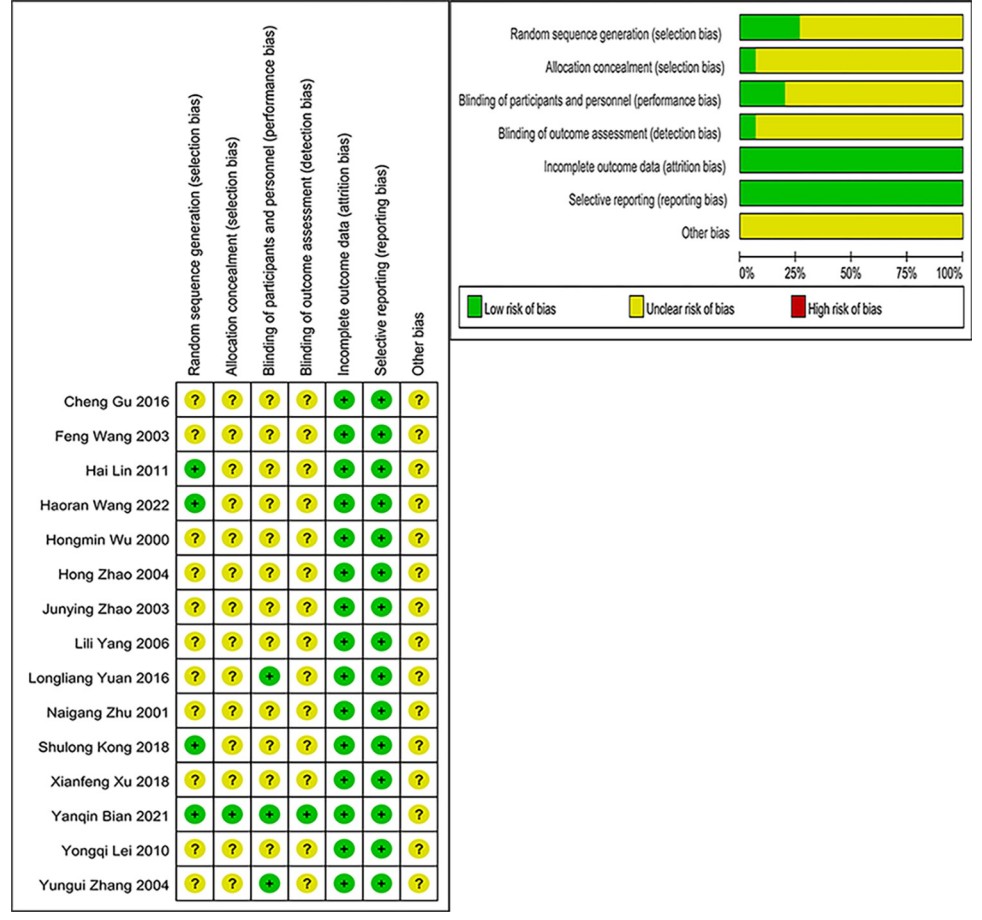

**Fig 2.** (a) Risk of bias summary. (b) Risk of bias graph.

heterogeneity. We excluded each study one by one to reanalyze and determine the robustness of the results. Based on the sensitivity analysis, the heterogeneity result was $I^2 = 0\%$ after excluding a study [28], so we used a fixed effects model to analyze. The differences between WFC and the control group remained statistically significant (RR 1.46; 95% CI 1.36, 1.57; P<0.00001). Notably, the overall effects (Z = 10.21, p<0.00001) still revealed that WFC was more effective than the control group (**Fig 4**). According to the sensitivity analysis result, this study [28] was most likely the underlying source of heterogeneity. Furthermore, WFC was still more effective in clinical efficacy than the control group after excluding other studies individually.

**Publication bias.** The funnel plot (**Fig 5**) assessed the fourteen studies' potential publication bias [21–33, 35]. According to the funnel plot, there existed potential publication bias. Thus, we excluded each study included in the analysis one by one and found that the two studies [28, 31] were more likely to have potential publication bias. Notably, the funnel plot excluded two studies [28, 31], as shown in **Fig 6**.

**Improvement of gastric mucosa under gastroscopy.** Four studies [21, 25, 26, 32] reported improved gastric mucosa under gastroscopy in CAG patients after treatment. A fixed effects model was conducted to analyze heterogeneity results ($I^2 = 0\%$). The differences between WFC and the control group in improving gastric mucosa under gastroscopy were

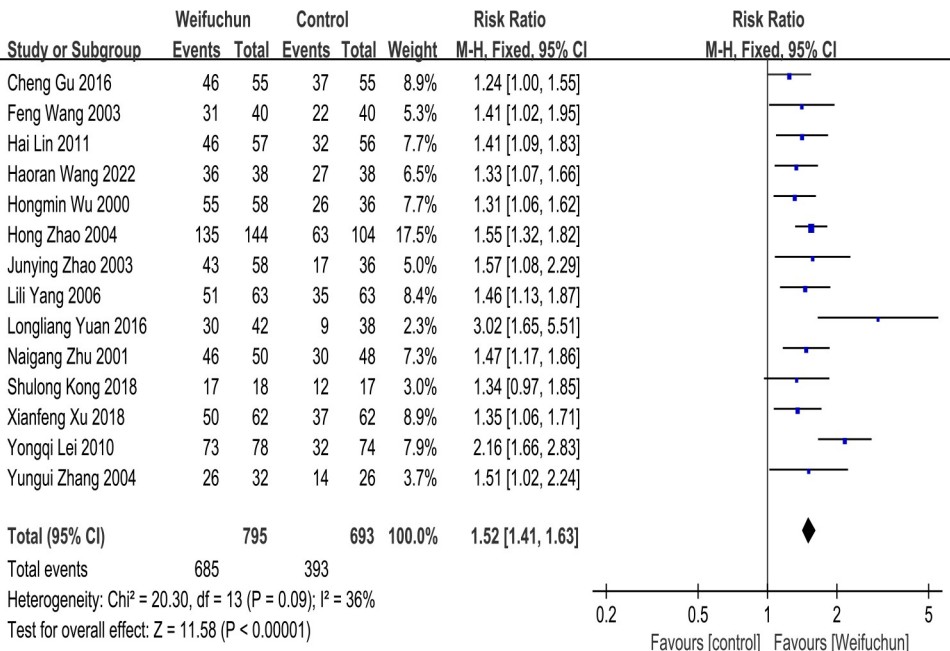

**Fig 3. Forest plot of clinical efficacy.**

statistically significant (RR 1.42; 95% CI 1.20, 1.68; P<0.0001). Furthermore, the overall effects (Z = 4.15, p<0.0001) indicated that WFC was more effective in improving gastric mucosa under gastroscopy than the control group (**Fig 7**).

**Improvement of histopathologic variations of gastric mucosa.** Four studies [21, 25, 33, 34] reported histopathologic variations of gastric mucosa, containing atrophy, IM, and

| Study or Subgroup | Weifuchun Events | Total | Control Events | Total | Weight | Risk Ratio M-H, Fixed, 95% CI | Risk Ratio M-H, Fixed, 95% CI |
|---|---|---|---|---|---|---|---|
| Cheng Gu 2016 | 46 | 55 | 37 | 55 | 9.6% | 1.24 [1.00, 1.55] | |
| Feng Wang 2003 | 31 | 40 | 22 | 40 | 5.7% | 1.41 [1.02, 1.95] | |
| Hai Lin 2011 | 46 | 57 | 32 | 56 | 8.4% | 1.41 [1.09, 1.83] | |
| Haoran Wang 2022 | 36 | 38 | 27 | 38 | 7.0% | 1.33 [1.07, 1.66] | |
| Hongmin Wu 2000 | 55 | 58 | 26 | 36 | 8.3% | 1.31 [1.06, 1.62] | |
| Hong Zhao 2004 | 135 | 144 | 63 | 104 | 19.0% | 1.55 [1.32, 1.82] | |
| Junying Zhao 2003 | 43 | 58 | 17 | 36 | 5.5% | 1.57 [1.08, 2.29] | |
| Lili Yang 2006 | 51 | 63 | 35 | 63 | 9.1% | 1.46 [1.13, 1.87] | |
| Longliang Yuan 2016 | 30 | 42 | 9 | 38 | 2.5% | 3.02 [1.65, 5.51] | |
| Naigang Zhu 2001 | 46 | 50 | 30 | 48 | 8.0% | 1.47 [1.17, 1.86] | |
| Shulong Kong 2018 | 17 | 18 | 12 | 17 | 3.2% | 1.34 [0.97, 1.85] | |
| Xianfeng Xu 2018 | 50 | 62 | 37 | 62 | 9.6% | 1.35 [1.06, 1.71] | |
| Yongqi Lei 2010 | 73 | 78 | 32 | 74 | | Not estimable | |
| Yungui Zhang 2004 | 26 | 32 | 14 | 26 | 4.0% | 1.51 [1.02, 2.24] | |
| | | | | | | | |
| Total (95% CI) | | 717 | | 619 | 100.0% | 1.46 [1.36, 1.57] | |
| Total events | 612 | | 361 | | | | |
| Heterogeneity: Chi² = 10.78, df = 12 (P = 0.55); I² = 0% | | | | | | | |
| Test for overall effect: Z = 10.21 (P < 0.00001) | | | | | | | |

**Fig 4. Forest plot of sensitivity analysis of clinical efficacy.**

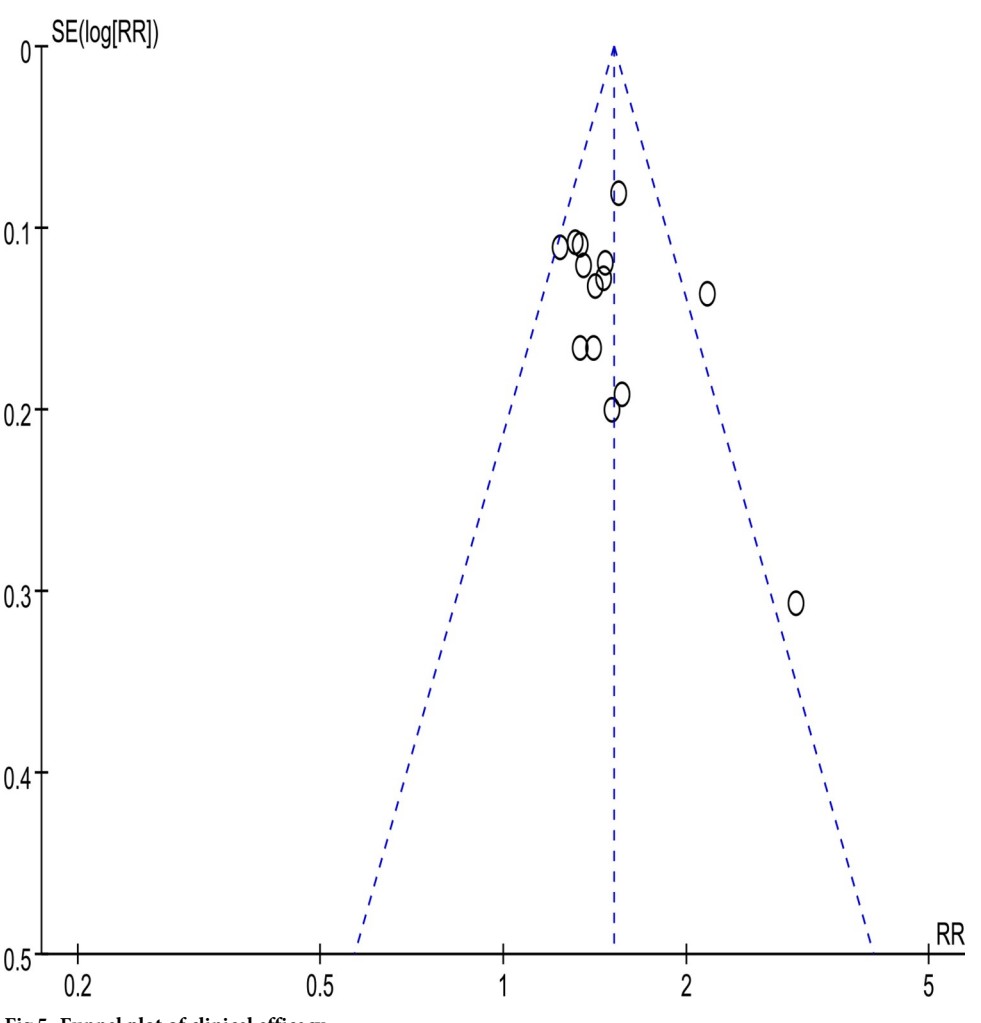

**Fig 5. Funnel plot of clinical efficacy.**

dysplasia. A random effects model was conducted to analyze heterogeneity results ($I^2$ = 74%). The differences between the WFC and control group in improving histopathologic variations of gastric mucosa were statistically significant (RR 2.34; 95% CI 1.22, 4.49; P = 0.01). Moreso, the overall effect (Z = 2.56, p = 0.01) showed that WFC was more effective than the control group in improving histopathologic variations of the gastric mucosa (**Fig 8**).

**H pylori inhibition rate.** Two studies [26, 32] reported the improvement of H pylori inhibition rate. A fixed effects model was performed to analyze heterogeneity results ($I^2$ = 0%). The differences between WFC and the control group in improving H pylori inhibition rate were statistically significant (RR 1.26; 95% CI 1.03, 1.55; P = 0.02). Furthermore, the overall effects (Z = 2.27, p = 0.02) revealed that WFC was more effective in improving H pylori inhibition rate than the control group (**Fig 9**).

**Safety evaluation.** We examined a safety evaluation of all enrolled studies. Eight studies reported adverse reactions in the treatment course [21, 22, 24, 25, 29–31, 33]. From these eight studies, there were no side effects noted in five studies[22, 24, 25, 29, 30], and three studies reported cases of adverse effects[21, 31, 33]. In contrast, other studies did not describe adverse effects. Moreso, no severe adverse events were reported in all enrolled studies. We used a fixed effects model to analyze heterogeneity results (P = 0.83 and $I^2$ = 0%). The differences between

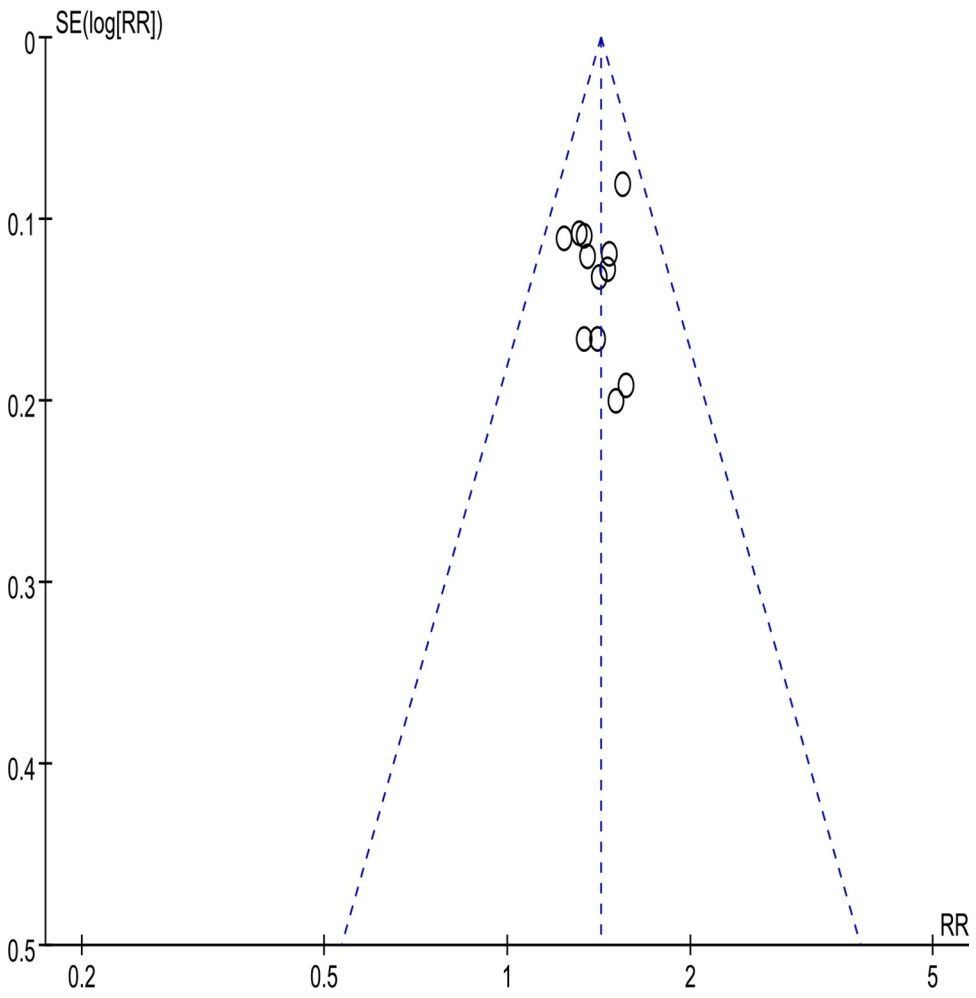

**Fig 6. Funnel plot of clinical efficacy after excluding two studies.**

the WFC and control group in safety assessment were statistically insignificant (RR 2.83; 95% CI 0.85, 9.38; P = 0.09). Furthermore, the overall effect (Z = 1.70, p = 0.09) indicated no significant differences between the two groups (**Fig 10**).

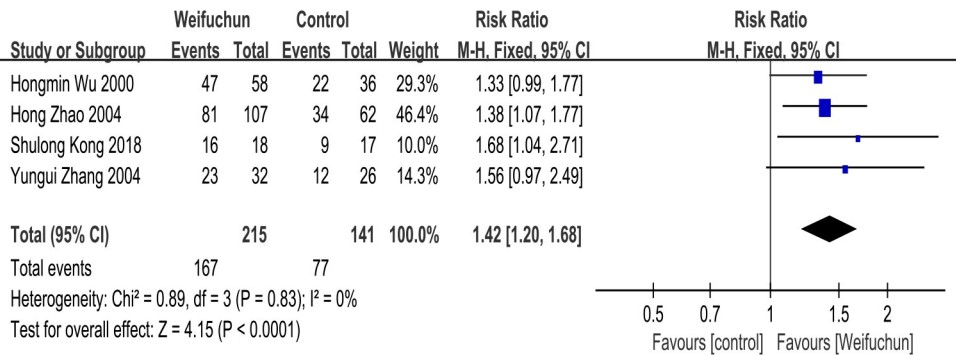

**Fig 7. Forest plot of improvement of gastric mucosa under gastroscopy.**

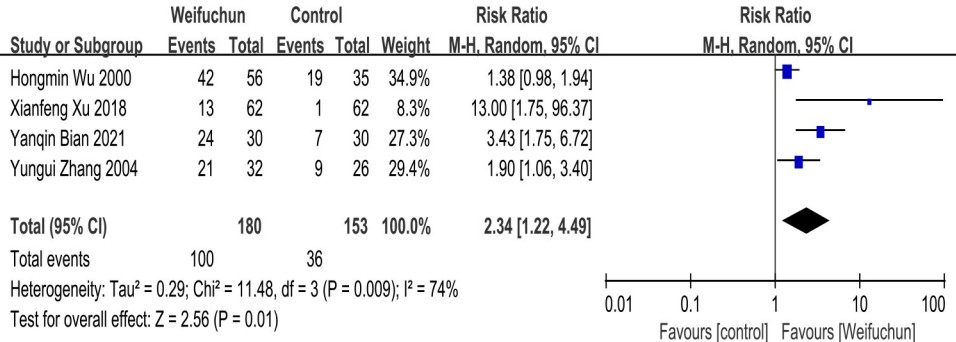

**Fig 8. Forest plot of improvement of histopathologic variations of the gastric mucosa.**

**Studies reporting drop outs and follow ups.** Only two studies [29, 34] reported cases dropped out during the treatment. In addition, none of the fifteen studies described followed up.

## Discussion

CAG is a significant premalignant lesion of gastric carcinoma. Thus, early attention and intervention to CAG are crucial to prevent progression to GC. Notably, WFC has been widely used for CAG in China. In this study, we conducted a meta-analysis to examine the effectiveness and safety of WFC for CAG. Our study revealed that: 1) WFC was more effective than the control group in clinical efficacy; 2) WFC showed more effects in improving H pylori inhibition rate, improving gastric mucosa under gastroscopy, and relieving histopathologic changes of the gastric mucosa.

Many studies [36–43] have shown the effectiveness of WFC for CAG patients, which is consistent with our results. Importantly, WFC significantly improved symptoms by inhibiting gastric acid secretion and regulating pepsinogen levels [44]. Additionally, WFC could suppress the expression of oncogenes and increase the expression of tumor suppressor genes to improve the degree of mucosal atrophy [45, 46]. Many studies have also shown that H pylori infection is closely related to atrophy, IM, and GC. By regulating the expression of pro-inflammatory and anti-inflammatory factors, WFC could reduce mucosal inflammation to improve atrophy and relieve symptoms [47]. WFC also had a specific inhibitory effect on H pylori [48]. In addition, the significant efficacy of WFC compared with the control group could be achieved by regulating specific signaling pathways such as the Hh-Wnt signaling pathway and NF-κB signaling pathway [47, 49]. According to the research of network pharmacology and molecular

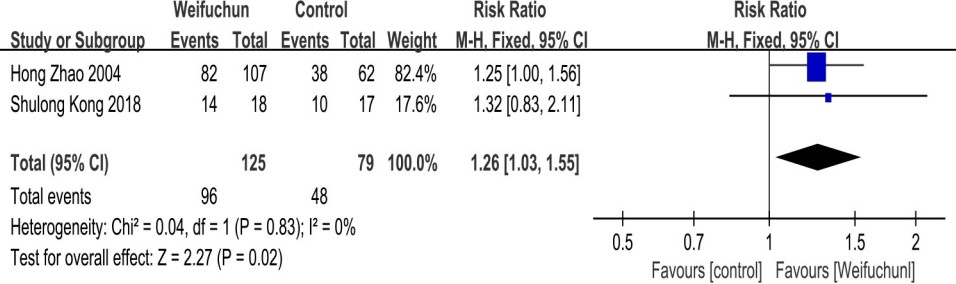

**Fig 9. Forest plot of H pylori inhibition rate.**

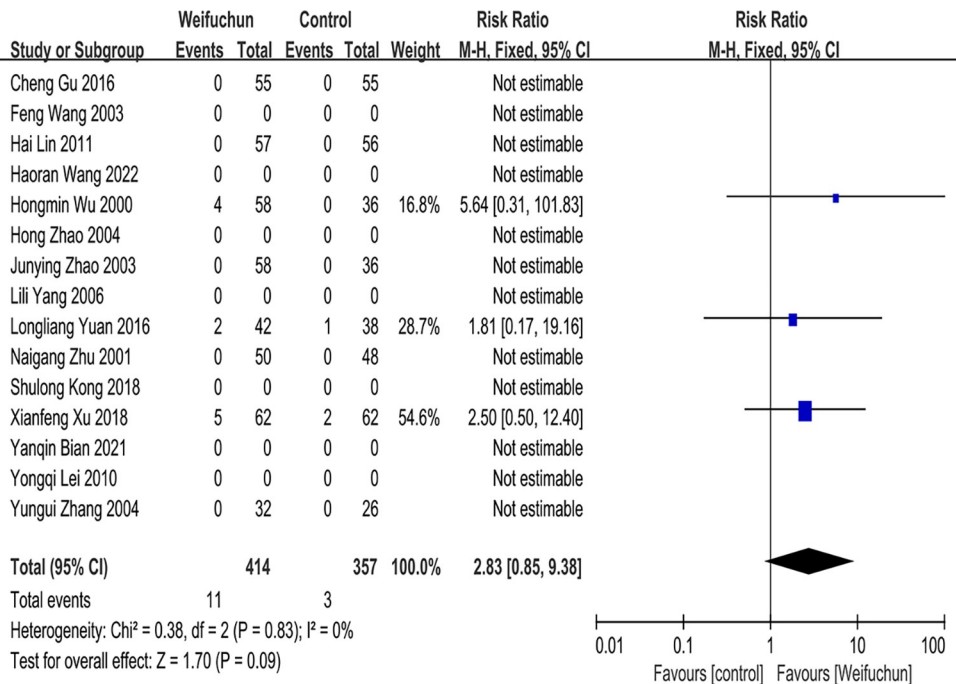

**Fig 10. Forest plot of safety evaluation.**

docking [50], WFC could play a role in treating CAG through multi-components, multi-targets and multi-pathways.

However, there still exists some inadequacy in this meta-analysis. Firstly, the limitations of methodological quality and research design could lead to a high-risk bias in all studies. Secondly, the duration of intervention was not identical in all enrolled studies, which might influence some results and result in clinical heterogeneity. Furthermore, no studies conducted a follow-up to determine the long-term efficacy of WFC.

## Conclusions

This study suggested that WFC was more effective than the control group in the effective clinical rate for CAG, and there was no significant difference in safety. However, more high-quality RCTs with standardized study and strict design should be conducted because of research design limitations in all enrolled studies.

## Supporting information

**S1 Checklist. PRISMA checklist.**
(PDF)

**S2 Checklist. Flow diagram of the study selection process.**
(PDF)

## Acknowledgments

The authors thank AiMi Academic Services (www.aimieditor.com) for English language editing and review services.

## Author Contributions

**Conceptualization:** Longhua Wang.

**Data curation:** Longhua Wang, Ping Li.

**Formal analysis:** Longhua Wang, Ping Li, Fenglei Wang.

**Funding acquisition:** Xia Ding.

**Investigation:** Longhua Wang, Shuying Ru, Lan Li.

**Methodology:** Longhua Wang, Xia Ding, Ping Li.

**Project administration:** Xia Ding, Fuwen Zhang.

**Resources:** Ping Li.

**Software:** Longhua Wang.

**Supervision:** Xia Ding, Fuwen Zhang.

**Validation:** Longhua Wang, Xia Ding, Shuying Ru.

**Visualization:** Longhua Wang, Ping Li, Fenglei Wang, Lan Li.

**Writing – original draft:** Longhua Wang.

**Writing – review & editing:** Longhua Wang, Xia Ding, Ping Li, Fuwen Zhang, Shuying Ru, Fenglei Wang, Lan Li.

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
