## [Decision Letter · Decision Letter 0]

14 Mar 2023

PONE-D-22-32660Efficacy and safety of Weifuchun tablet for chronic atrophic gastritis: A systematic review and meta-analysisPLOS ONE

Dear Dr. wang,

Thank you for submitting your manuscript to PLOS ONE. After careful consideration, we feel that it has merit but does not fully meet PLOS ONE’s publication criteria as it currently stands. Therefore, we invite you to submit a revised version of the manuscript that addresses the points raised during the review process.

We look forward to receiving your revised manuscript.

Kind regards,

Mohamed Ezzat Abd El-Hack

Academic Editor

PLOS ONE

Journal Requirements:

2. PLOS ONE does not copy edit accepted manuscripts (https://journals.plos.org/plosone/s/criteria-for-publication#loc-5). To that effect, please ensure that your submission is free of typos and grammatical errors.

3. We note that this manuscript is a systematic review or meta-analysis; our author guidelines therefore require that you use PRISMA guidance to help improve reporting quality of this type of study. Please upload copies of the completed PRISMA checklist as Supporting Information with a file name “PRISMA checklist”.

Reviewers' comments:

Reviewer's Responses to Questions

**Comments to the Author**

1. Is the manuscript technically sound, and do the data support the conclusions?

Reviewer #1: Yes

Reviewer #2: Yes

2. Has the statistical analysis been performed appropriately and rigorously? 

Reviewer #1: Yes

Reviewer #2: Yes

3. Have the authors made all data underlying the findings in their manuscript fully available?

Reviewer #1: Yes

Reviewer #2: Yes

4. Is the manuscript presented in an intelligible fashion and written in standard English?

Reviewer #1: Yes

Reviewer #2: Yes

5. Review Comments to the Author

Reviewer #1: The meta-analysis revealed that there had a significant statistic difference in effectiveness compared Weifuchun with control group, but there was no significant difference in safety. More high-quality clinical studies are needed in the future. nice work by the authors

Reviewer #2: The valuable study was well-designed and conducted, and the authors reported methods and results clearly and appropriately. However, some minor concerns need to be addressed as follows:1) Abstract Line 30-31, what does “Chinese patent medicine + western medicine” mean? A more precise description is expected.2) Introduction section: the necessity of performing this study should be highlighted and elaborated. 3)Methods section: Lines 96, 104, 107, 117, and 125, the name of researchers should be provided; 3) Discussion section, the authors should interpret the results further, place them in the context of currently available findings, explain in-depth why does Weifuchun tablet work better for ACG? and 4) The manuscript needs extensive revision for language and grammar.

6. PLOS authors have the option to publish the peer review history of their article (what does this mean?). If published, this will include your full peer review and any attached files.

Reviewer #1: **Yes: **PUGAZHENTHAN THANGARAJU

Reviewer #2: No

---

## [Author Response · Author response to Decision Letter 0]

25 Mar 2023

Rebuttal Letter

Point-by-point responses to editor and reviewers’ comments.

Journal Requirements:

Response 1: Thanks for your reminding. We have downloaded the PLOS ONE style templates and fully addressed the format carefully.

2. PLOS ONE does not copy edit accepted manuscripts (https://journals.plos.org/plosone/s/criteria-for-publication#loc-5). To that effect, please ensure that your submission is free of typos and grammatical errors.

Response 2: Thank you for reminding. We revised the whole manuscript carefully to avoid language errors. In addition, we consulted a professional editing service to check the English.

3. We note that this manuscript is a systematic review or meta-analysis; our author guidelines therefore require that you use PRISMA guidance to help improve reporting quality of this type of study. Please upload copies of the completed PRISMA checklist as Supporting Information with a file name “PRISMA checklist”.

Response 3: Thank you for your suggestion. We have uploaded an update PRISMA checklist as supporting information with a file according to the revised manuscript.

Response 4: Thanks for your reminding. We have included captions for supporting information files at the end of our manuscript.

Important: If there are ethical or legal restrictions to sharing your data publicly, please explain these restrictions in detail. Please see our guidelines for more information on what we consider unacceptable restrictions to publicly sharing data:http://journals.plos.org/plosone/s/data-availability#loc-unacceptable-data-access-restrictions. Note that it is not acceptable for the authors to be the sole named individuals responsible for ensuring data access. 

Response 5: Thank you for your suggestion! We have uploaded our study’s dataset to Figshare. https://doi.org/10.6084/m9.figshare.22298005.v2

Response 6: Thanks for your reminding. We have reviewed our reference list carefully to ensure that it is complete and correct. According to Reviewer #2 comments, we added a reference in the revised manuscript( Lines 464-467).

Reviewers' comments:

Reviewer #1: The meta-analysis revealed that there had a significant statistic difference in effectiveness compared Weifuchun with control group, but there was no significant difference in safety. More high-quality clinical studies are needed in the future. nice work by the authors

Response: Thanks for the reviewer’s positive comments.

Reviewer #2: The valuable study was well-designed and conducted, and the authors reported methods and results clearly and appropriately. However, some minor concerns need to be addressed as follows:1) Abstract Line 30-31, what does “Chinese patent medicine + western medicine” mean? A more precise description is expected.2) Introduction section: the necessity of performing this study should be highlighted and elaborated. 3)Methods section: Lines 96, 104, 107, 117, and 125, the name of researchers should be provided; 3) Discussion section, the authors should interpret the results further, place them in the context of currently available findings, explain in-depth why does Weifuchun tablet work better for ACG? and 4) The manuscript needs extensive revision for language and grammar.

Response: Thanks for the reviewer’s positive and constructive comments and suggestions. Point-by-point responses as follows:

Comment 1: Abstract Line 30-31, what does “Chinese patent medicine + western medicine” mean? A more precise description is expected.

Response 1: Thank you for your suggestion! We have changed the description to “Western medicine and other Chinese patent medicine” to avoid confusing.(Line 31)

Comment 2: Introduction section: the necessity of performing this study should be highlighted and elaborated.

Response 2: Thanks for your suggestion! We have highlighted the necessity of performing this study in introduction section( Line 70-71).

Comment 3: Methods section: Lines 96, 104, 107, 117, and 125, the name of researchers should be provided.

Response 3: Thanks for your advice! We have added the name of researchers in methods section.(Lines 82, 100, 108, 111, 121-122, 126, and 131)

Comment 4: Discussion section, the authors should interpret the results further, place them in the context of currently available findings, explain in-depth why does Weifuchun tablet work better for ACG?

Response 4: Thank you for your suggestion! We have further explained and elaborated why WFC was better for CAG in discussion section ( Lines 266-278).

Comment 5: The manuscript needs extensive revision for language and grammar.

Response 5: Thanks for your suggestion! We revised the whole manuscript carefully to avoid language errors. In addition, we consulted a professional editing service to check the English.

---

## [Editor Report · Decision Letter 1]

30 Mar 2023

Efficacy and safety of Weifuchun tablet for chronic atrophic gastritis: A systematic review and meta-analysis

PONE-D-22-32660R1

Dear Dr. wang,

We’re pleased to inform you that your manuscript has been judged scientifically suitable for publication and will be formally accepted for publication once it meets all outstanding technical requirements.

Kind regards,

Mohamed Ezzat Abd El-Hack

Academic Editor

PLOS ONE

---

## [Editor Report · Acceptance letter]

5 Apr 2023

PONE-D-22-32660R1 

Efficacy and safety of Weifuchun tablet for chronic atrophic gastritis: A systematic review and meta-analysis 

Dear Dr. Wang:

I'm pleased to inform you that your manuscript has been deemed suitable for publication in PLOS ONE. Congratulations! Your manuscript is now with our production department. 

Kind regards, 

on behalf of

Dr Mohamed Ezzat Abd El-Hack 

Academic Editor

PLOS ONE